# Proactive Scheduling for Job-Shop Based on Abnormal Event Monitoring of Workpieces and Remaining Useful Life Prediction of Tools in Wisdom Manufacturing Workshop

**DOI:** 10.3390/s19235254

**Published:** 2019-11-29

**Authors:** Cunji Zhang, Xifan Yao, Wei Tan, Yue Zhang, Fudong Zhang

**Affiliations:** 1Yichang Key Laboratory of Robot and Intelligent System, China Three Gorges University, Yichang 443002, China; 2School of Mechanical and Power Engineering, China Three Gorges University, Yichang 443002, China; 3School of Mechanical and Automotive Engineering, South China University of Technology, Guangzhou 510640, China; 4School of Computer Science and Technology, Dongguan University of Technology, Dongguan 523808, China; tanwphmr@163.com; 5School of Mechanical Engineering, Chongqing University, Chongqing 401331, China; zhangyuejxdz@163.com; 6School of Mechanical Engineering, Tianjin University of Technology, Tianjin 300384, China; zhangfudongqwe@163.com

**Keywords:** proactive scheduling, abnormal event monitoring, RFID, remaining useful life, wireless accelerometer, wisdom manufacturing

## Abstract

The job-shop scheduling is an important approach to manufacturing enterprises to improve response speed, reduce cost, and improve service. Proactive scheduling for job-shop based on abnormal event monitoring of workpieces and remaining useful life prediction of tools is proposed with radio frequency identification (RFID) and wireless accelerometer in this paper. Firstly, the perception environment of machining job is constructed, the mathematical model of job-shop scheduling is built, the framework of proactive scheduling is put forward, and the hybrid rescheduling strategy based on real-time events and predicted events is adopted. Then, the multi-objective, double-encoding, double-evolving, and double-decoding genetic algorithm (MD3GA) is used to reschedule. Finally, an actual prototype platform to machine job is built to verify the proposed scheduling method. It is shown that the proposed method solves the integration problem of dynamic scheduling and proactive scheduling of processing workpieces, reduces the waste of redundant time for the scheduling, and avoids the adverse impact on abnormal disturbances.

## 1. Introduction

As an important part of a manufacturing enterprise, the production workshop is the intersection of information flow and logistics in the production process. Production information includes processing data, production management, logistics information, production scheduling, human resources, and other heterogeneous information. The job-shop scheduling is an important method of manufacturing enterprises to improve response speed, reduce cost, and improve service. Nowadays, the manufacturing workshop is a complex system, and with the development of personalization and small batch production model, the traditional scheduling methods are no longer able to adapt to the new production model of personalized customization. The production plan is adjusted online according to real-time information such as customer requirements, product orders, raw materials inventory, equipment status, and product inventory. Specifically, the equipment is scheduled and controlled proactively according to real-time or predicted events in the manufacturing workshop, which takes on important and practical significances to improve the productivity and competitiveness in manufacturing enterprises.

The rapid development of information technology promotes the continuous upgrading of manufacturing mode. Especially, wisdom manufacturing (WM) was proposed recently [1], which changes the essence of manufacturing mode. Internet of things (IoT) is the key technology of humans–computers–things interconnection of WM, which applies a knowledge network to process sensing data collected by the IoT. WM is a fusion of cloud manufacturing (CM), internet of manufacturing things (IoMT), semantic network manufacturing, and enterprise 2.0, where various manufacturing resources are connected and formed a unified resource pool under the support of IoT, internet of knowledge (IoK), internet of services (IoS), internet by and for people (IbfP), etc. Smart response is made by the co-determination of humans, computers, and things according to the customer’s personalized requirements and context awareness. Manufacturing services that are customized, on-demand, proactive, transparent, and trusted are provided for customers in the whole manufacturing life cycle.

With the development of market globalization, facing the growing customers and production demands of personalization, customization and just-in-time, uncertainties such as dynamic and changeful orders, shortage of raw materials, advance of delivery, and machine faults that will happen to manufacturing enterprises, agility, robustness, and initiative are needed for the manufacturing system to face the more complex challenges [2]. Deep fusion of information technology and manufacturing technology promotes manufacturing modes to upgrade constantly and based on proactive computing [3] and big data technologies, proactive manufacturing (PM) is proposed, which is a big-data-driven emerging manufacturing paradigm [4]. It is essentially a kind of WM based on big data, which realizes the humans–computers–things collaboration with perception, analysis, orientation, decision, adjustment, and control, and provides customers with customized/personalized products and services. Static scheduling (SS) is considered as a non-deterministic polynomial (NP) problem of production manufacturing, however the scheduling system often runs in high dynamic and uncertainty; dynamic scheduling (DS) emerges at the right moment due to the dynamic manufacturing environment, and forms an NP problem more complex than SS [5]. In order to solve the uncertain problems, these scheduling methods such as real-time scheduling, reactive scheduling, rescheduling, online scheduling, and adaptive scheduling are adopted in production scheduling [6]. However, these scheduling methods make immediate and dynamic reaction after abnormal disturbances happen in the manufacturing system, which causes damage to the manufacturing system or a waste of scheduling redundant time. The disturbance events of possible occurrence are predicted in advance before the occurrence of abnormal disturbances in proactive scheduling (PS), so as to optimize the prior allocation of manufacturing resources and autonomously adapt to the changes of the manufacturing environment, aim to improve production efficiency, and reduce production cost. WM/PM is aimed at providing customers with personalized and customized products.

There are a wide range of uncertainties of the actual product manufacturing, such as new order insertion, raw material shortage, machine failure, and tool wear, etc., and a number of performance requirements need to be considered in the production, such as maximum completion time, delivery time, production cost, and inventory, etc., and some requirements may conflict. The actual manufacturing workshop is a process combining static, dynamic, and proactive scheduling. At present, there are many different classification methods of the scheduling model of manufacturing workshop. The application of these models is directly related to a problem, and when the practical problems of the workshop scheduling can be solved, and the scheduling model is closed to the actual production status, the model runs better. An intuitive classification method of production scheduling models is shown in Figure 1, which classifies scheduling problems according to dynamics and complexity. Therefore, proactive scheduling is a further development of real-time and dynamic scheduling and is a more complex scheduling problem.

The rest of this paper is organized as follows: Section 2 reviews the literature of dynamic scheduling and proactive scheduling in machining, states the current problem of scheduling. Section 3 introduces the proactive scheduling scheme in detail. Section 4 builds an actual prototype platform for a machining job to verify the proposed scheduling method. Conclusions and suggestions for future work are given in Section 5.

## 2. Literature Review

The scheduling scheme is modified or rescheduled in the case of high dynamics and occurrence of abnormal events during the dynamic scheduling. The occurrence of abnormal events is predicted in advance based on real-time and historical data onto the production site during the proactive scheduling, so as to reallocate manufacturing system resources. The goal is to minimize the impact on disturbances on scheduling performance. At present, many scholars have conducted research on dynamic scheduling. For example, Zheng et al. proposed a dynamic scheduling method based on neuroendocrine regulation mechanism, which flexibly responds to uncertain disturbances in production workshop based on the principle of hormone regulation [2]. Based on genetic algorithm and tabu search algorithm, Zhang et al. solved the problem of arbitrary workpiece insertion and machine fault with the workshop [5]. Zakaria et al. solved the problem of new order insertion based on genetic algorithm; the rescheduling method matched the recombination and non-recombination strategies and adjusted new orders according to the idle time of the machine [6]. Vieira et al. proposed a new rescheduling system, which described rescheduling strategies and methods [7]. Umar et al. applied hybrid multi-objective genetic algorithm, aiming at optimizing the maximum completion time, traveling time of an automatic guided vehicle (AGV), and delay penalty, and the optimal scheme for scheduling and path selection was generated [8]. Dong et al. implemented integrated scheduling for processing machines and AGV with an improved genetic algorithm [9]. Using a heuristic algorithm and genetic algorithm, Rahman et al. solved the problem of order receiving in permutation flow-shop scheduling (PFS) [10]. Li et al. proposed a discrete optimization algorithm based on teaching learning (TL) to solve the problem of job-shop rescheduling, which embedded an improved iterative greedy local search algorithm to improve the searchability of teaching-learning algorithm [11]. Liu et al. proposed a multi-objective flexible dynamic scheduling algorithm based on adaptive genetic algorithm, which achieved the real-time scheduling function of machine fault, processing task changing, and periodic rescheduling based on a hybrid rescheduling strategy driven by events and cycles [12]. Li et al. designed a flexible job-shop scheduling method of uncertain environment, which can adapt to three kinds of disturbances such as order transaction, operation delay, and machine fault [13]. Setiawan et al. developed a scheduling algorithm for a flexible manufacturing system, which takes into account the factors of tool fault and tool life [14]. Rokni et al. adopted the Pareto-optimality concept combined with fuzzy set theory to optimize the pipe spool fabrication shop scheduling [15]. Taghaddos et al. proposed the simulation-based auction protocol (SBAP) to realize the effective allocation of resources and satisfaction of various constraints [16]. In terms of proactive scheduling, Wang et al. proposed a knowledge-based proactive scheduling method, which applies the multi-objective evolutionary algorithm of elite non-dominant sorting to solve the problems of machine failure and degradation [17]. Rahmani provided a proactive-reactive approach to a two-machine flow shop system, considering uncertain processing time and unexpected machine failure [18]. Cui et al. dealt with the integration of production scheduling and maintenance planning through optimizing the bi-objective of quality robustness and solution robustness for flow shops [19].

Most research on production scheduling focuses on isolated scheduling schemes, and the scheduling optimization algorithms mostly adopt a heuristic algorithm or a hybrid intelligent algorithm. The occurrence of disturbances is assumed randomly, and scheduling schemes for different disturbances are obtained through abstract simplification, which are not verified in actual production workshop. However, in the actual environment of manufacturing, the disturbances often occur randomly, and it is not just a scheduling scheme to deal with abnormal events. Different scheduling schemes need to be adopted according to the real-time monitoring in the manufacturing workshop and the disturbances predicted through the monitored data. Based on the wisdom manufacturing mode, which is put forward, the abnormal events are monitored by RFID in the bottom layer of a physical production system and a wireless accelerometer is used to monitor tool vibration and predict tool life. This paper focuses on the proactive scheduling scheme for job-shop based on abnormal event monitoring of workpieces and remaining useful life prediction of tools, which contains dynamic scheduling to respond to abnormal conditions in real-time and proactive scheduling driven by predicted events, and the real job-shop experiment platform is built to validate the scheme.

## 3. Proactive Scheduling Scheme

In this section, the perception environment of a machining job is constructed, the mathematical model of job-shop scheduling is built, the framework of proactive scheduling is put forward, the scheduling strategy is discussed, and finally, the multi-objective, double-encoding, double-evolving, and double-decoding genetic algorithm (MD3GA) is introduced in detail.

### 3.1. Perceptual Environment Construction

Machining situations of workpieces and abnormal disturbances need to be perceived in time during the machining process of workpieces, and the impending disturbances are predicted. A perceived environment of machining job is built as shown in Figure 2. The entire production workshop is covered by the network (e.g., local area network (LAN), wireless fidelity (WiFi), blue tooth, etc.) in this environment; the production workshop consists of a dispatching center, raw material and product warehouse (*N_rpw_*), AGV, numerical control (NC) machine tool (*N_ncmx_*), etc. Directional ultra-high-frequency (UHF) readers equipped with LAN (WiFi) are installed on each workstation and anti-metal ceramic RFID tags are pasted on every workpiece. Raw materials or semi-finished products are distributed to the processing stations by AGV and the whole process from raw material to product is monitored. Each workstation is equipped with RFID sensing nodes, which sense identification (ID) number, time, location, and other data of arrived workpieces in real time; the abnormal disturbance events are mined in real time using complex event processing (CEP) technology [20], and formed the real-time state matrix *S* = (*α_i_*_j_) (1 ≤ *i* ≤ *m*, 1 ≤ *j* ≤ 4).
(1)$$S=\left(a_{ij}\right)=\left[\begin{array}{c} \begin{array}{cccc} a_{11} & a_{12} & a_{13} & a_{14}\\ a_{21} & a_{22} & a_{23} & a_{24}\\ a_{31} & a_{32} & a_{33} & a_{34}\\ \vdots & \vdots & \vdots & \vdots \\ a_{m1} & a_{m2} & a_{m3} & a_{m4} \end{array} \end{array}\right]$$
where *i* is the serial number of the workpiece and *j* is the serial number of the workstation; if *α_ij_* is equal to 1, the processing situation of the workpiece is normal; if *α_ij_* is equal to 0, the processing situation of the workpiece is abnormal.

Wireless triaxial accelerometers are adopted to monitor vibration signal of tools for the machining process of workpieces and implement remaining useful life prediction. For example, the workpiece is machined by milling machine (model: Xendoll Tech C000017); the milling cutter adopts double-edged cutter of micro particles and tungsten carbide (model: Seco S550, diameter: 6 mm); a wireless triaxial accelerometer (model: M69) is used to measure the vibration of milling process; and the matching wireless base station (model: M90) is used to regulate the vibration signal and transmit to the dispatching center. The sampling frequency of the triaxial accelerometer is 1 kHz/channel, and a portable digital microscope (model: MSUSB401) is used to measure tool wear after milling. The vibration data collected are input into the neural fuzzy network (NFN) after de-noising, feature extraction, and feature selection [21], and aim to realize the prediction of tool remaining useful life. The remaining useful life of each machine tool forms a life status matrix *W*.
(2)$$W=\left[\begin{array}{cccc} w1 & w2 & w3 & w4 \end{array}\right]$$
where *w*1 is the tool remaining useful life of the first machine tool (here expressed by the number of workpieces that the tool can also machine), and so on.

### 3.2. Proactive Scheduling Mathematical Model

The scheduling problem of wisdom manufacturing workshop is described as *n* workpieces {*J*_1_, *J*_2_, …, *J_n_*} that are machined in *m* machines {*M*_1_, *M*_2_, …, *M**_m_*}; each workpiece is machined with *h* processes and each process can be machined by any one of multiple machine tools; the processing path of *h* process for *n* workpieces may be different. *O_ij_* represents the processing operation of the *j^th^* process for the *i^th^* workpiece, and *T_ij_* represents the processing time of the *j^th^* process for the *i^th^* workpiece. Then, the completion time of the *i^th^* workpiece is:(3)$$C_{i}=\max_{1\leq j\leq h}\left(T_{ij}\right).$$

When the initial scheduling scheme is affected by real-time and predicted disturbance events, the initial scheduling scheme is slightly modified to form the proactive scheduling (dynamic scheduling) scheme. The performance robustness measure of the scheduling scheme is:(4)$$r_{p}=\left(\frac{\left|C_{\max}^{p}-C_{\max}^{b}\right|}{C_{\max}^{b}}\right)\times 100$$
where $C_{\max}^{p}$ is the processing completion time of the workpiece in the proactive scheduling (dynamic scheduling) scheme and $C_{\max}^{b}$ is the processing completion time of the workpiece in the initial scheduling scheme.

The stability robustness measure of the initial scheduling scheme is:(5)$$r_{s}=\frac{\sum_{i=1}^{n}\sum_{j=1}^{h}\left|C_{ij}^{p}-C_{ij}^{b}\right|}{\sum_{i=1}^{n}\sum_{j=1}^{h}O_{ij}}$$
where $C_{ij}^{p}$ is the processing completion time of the *j^th^* process of the *i^th^* workpiece in the proactive scheduling (dynamic scheduling) scheme and $C_{ij}^{b}$ is the processing completion time of the *j^th^* process for the *i^th^* workpiece in the initial scheduling scheme.

When real-time and predicted events affect the initial scheduling scheme, the corresponding proactive scheduling (dynamic scheduling) scheme is formed, aiming to minimize the impact of disturbance events on the initial scheduling scheme. The robustness measure of the scheduling scheme is expressed as:(6)$$r=r_{p}+r_{s}.$$

The constraints on the scheduling problem are as follows:The processing path of each workpiece may be different;At each moment, each machine tool can only be used to machine one process, and the process is not allowed to be interrupted; each machine tool is equipped with input/output buffer;Only one machine tool can be selected for each process;The processing time of each process has been determined;A workpiece cannot be processed on different machine tools at the same time;The preparation time of the process is ignored, or contained in the processing time;In case of real-time disturbance or predicted event, the machining process without impact will continue to machine until the process is completed.

### 3.3. Proactive Scheduling Framework

The proactive scheduling framework of machining jobs is shown in Figure 3, which contains a dynamic scheduling scheme. The disturbance is inevitable during the manufacturing process; when the abnormal disturbance occurs in the manufacturing workshop, the disturbance type (such as buffer blocking, etc.) is judged at first, and then the affected workpieces, processes, and machines are determined and the scheduling parameters are updated to generate a new dynamic scheduling scheme; at this time, the scheduling scheme is executed. In order to avoid the waste of redundant time in the scheduling and reduce the loss (sometimes serious) of the manufacturing system caused by abnormal disturbance, before the abnormal disturbances occur, the possible disturbances (such as exhausted tool life, etc.) in the future are predicted according to the real-time data and historical data during the machining and necessary preventive maintenance measures (such as replacement of tools, etc.) are taken to avoid the occurrence of disturbance. Then, the affected workpieces and processes are determined and the scheduling parameters are updated to generate a new proactive scheduling scheme. The goal of proactive scheduling is to minimize the impact on the initial (previous) scheduling scheme caused by abnormal disturbances [22].

### 3.4. Proactive Scheduling Strategy

Rolling window is an important method to solve the problem of scheduling [23]. For dynamic scheduling, the uncertain scheduling problem is decomposed into a series of dynamic but certain problems, and the dynamic scheduling process is decomposed into a series of continuous and static scheduling intervals; then, each scheduling interval is optimized by the scheduling algorithm and the system is optimal at each scheduling interval. The main idea of rolling window is that a workpiece window is adopted to realize a rolling scheduling, some workpieces are selected to add to the workpiece window in the initial scheduling, and the optimization algorithm is adopted to optimize and generate the initial scheduling scheme. When the disturbances occur in the machining process, the initial scheduling scheme is no longer adapted to the environment; then, the dynamic scheduling scheme restarts, the completed workpieces are removed from the workpiece window in the dynamic scheduling, the workpieces waiting for machining are added to the workpiece window, the workpieces in current window are optimized again, and the process is repeated until all workpieces are finished to machine.

Rolling scheduling strategy contains a rescheduling mechanism and a workpiece window. The rescheduling mechanism contains period-driven rescheduling, event-driven rescheduling, and period-event-driven rescheduling. Rescheduling is executed at the interval of the production cycle in period-driven rescheduling, and this mechanism can better maintain the stability of workshop production. Event-driven rescheduling means that when a real-time disturbance event occurs to the system, the rescheduling starts immediately; this mechanism can well respond to real-time events. Period-event-driven rescheduling integrates the advantages of the above two mechanisms, which can not only respond to the disturbance events in the manufacturing workshop well, but also maintain stable production. However, none of the three methods have the ability to predict future disturbance events.

Proactive scheduling adopts event-driven rescheduling based on predicted disturbances, the disturbance events (such as tool life exhausted event, etc.) are predicted according to real-time data and historical data collected in the manufacturing workshop before the disturbance events occur; some measures (such as replacing the tool) are implemented in advance. In this paper, aiming at real-time disturbance events detected by RFID and tool remaining useful life events detected and predicted by a wireless accelerometer, rescheduling based on real-time events and predicted events is adopted to realize the fusion of dynamic scheduling and proactive scheduling in the machining workshop.

A workpiece window needs to be defined in the rolling scheduling strategy; at the time of rescheduling, only the workpieces in the workpiece window at that time are scheduled. The diagram of the workpiece window for rolling scheduling is shown in Figure 4. The workpieces in the window are divided into a completed workpiece set, processing workpiece set, unprocessed workpiece set, and waiting workpiece set. The unprocessed workpiece set refers to the workpieces that have been scheduled and not been machined; the waiting workpiece set refers to the workpieces that have not been scheduled and are waiting to be scheduled. At the time of rescheduling in the rolling scheduling strategy, it is only necessary to remove the completed workpiece set from the current workpiece window, and then add the waiting workpiece set. Finally, the workpieces in the workpiece window are scheduled by the corresponding optimization algorithm.

### 3.5. Proactive Scheduling Algorithm

Genetic algorithm is rooted in the mechanisms of evolution and natural genetics [24]. Individuals best suited to competition for scanty resources survive, it is essential for the survival of individuals to adapt to a changing environment. The various features that uniquely characterize an individual determine its survival capacity; the features, in turn, are determined by the individual’s genetic content. It presumes that the potential solution of a problem is an individual and can be represented by a set of features. Each feature is controlled by a basic unit called a gene. The sets of genes controlling features form the chromosomes. Competition among individuals for scant resources such as food and space and for mates results in the fittest individuals dominating over weaker ones [25]. The fittest individuals survive and reproduce, a natural phenomenon called “the survival of the fittest.” New combinations of genes are generated from patents through crossover, and mutation causes sporadic and random alteration of the bits of strings. Throughout a genetic evolution, the fittest chromosome has the tendency to yield good-quality offspring, which means a better solution to the problem. The genetic algorithm has a stronger ability to solve nonlinear optimization problem, where each chromosome is used to represent an optimal solution to a problem; a chromosome can easily express a potential solution to a simple problem, but it is difficult to accurately express complex problem solutions (such as the machining process of a workpiece and machine encoding) at the same time, in order to solve the integrated scheduling problem of the machines and AGV and deal with the impact of the real-time and predicted disturbance events for the initial scheduling scheme. In this paper, an improved multi-objective, double-encoding, double-evolving, and double-decoding genetic algorithm (MD3GA) is adopted. The flowchart is shown in Figure 5. The double-encoding method divides each chromosome code into two layers (process code and machine code), which uses one chromosome to express the solution to a complex problem accurately. The chromosome is evolved by double-evolving method, and in the first evolution, the minimum completion time is the target to form the initial scheduling scheme, and the minimum completion time and robustness measure (weight sum) is the target in the second evolution to form the proactive scheduling scheme considering disturbance events. The decoding adopts a double-decoding method. In the first decoding, the processing order, processing machine, start processing time, and end time of the workpiece without considering the AGV distribution time are obtained. The second decoding takes into account the AGV distribution time to obtain the specific processing time (including distribution time) of each process.

#### 3.5.1. Double-Encoding

It is fundamental to the genetic algorithm that the optimization problem’s variables are represented by the encoding mechanism, and the encoding mechanism depends on the nature of the problem variables. Many optimization problems have real-valued continuous variables. As a way of chromosome encoding, integer encoding is used to represent the machining process of the workpiece and machine assignment information on each process at the same time. A chromosome is encoded by double-encoding method; the first layer is the process code of the machining workpiece, namely the process code, and the second layer is the machine code based on the available machine selection for each process, known as machine code. The total number of machining workpieces are *n*, the process amount of each workpiece is *h*, and the encoding length of each chromosome is 2∗*n*∗*h*, where the first *n*∗*h* bits of the chromosome represents the processing sequence of all workpieces on the machines and the last *n*∗*h* bits represents the processing machine number selected by each process of the workpiece. The processing sequence of six workpieces on four machines is expressed by the chromosome in Figure 6; every workpiece is machined from three processes and each process is performed on a machine selected from a different number of machines. Eighteen processes of six workpieces are in the first part of the chromosome and the machine serial number of every process is in the second half part; for example, the first process of the sixth workpiece is machined in the first (*M*_2_) optional machine and the second process is machined on the first (*M*_1_) optional machine.

#### 3.5.2. Fitness Function Calculation

The objective function provides the mechanism for evaluating each chromosome, and the fitness function is used to represent the fitness of an individual chromosome. In this paper, double-evolving method is adopted to correspond to different fitness functions of each level. The fitness function of the first-level evolutionary chromosome is the completion time of the workpiece, and its target is to minimize the maximum completion time, namely:(7)$$f_{1}=\min\left(\underset{1\leq i\leq n}{\max}(C_{i})\right)$$
where *C_i_* is the completion time of the *i^th^* workpiece (*i* = 1, 2, …, *n*). In order to calculate the individual fitness function, the chromosome needs to be completely reduced into machining sequence and machine serial number of each process for the workpiece. When calculating the completion time of the *j^th^* process for the *i^th^* workpiece, by comparing the start time of this process and the end time of the (*j* − 1)*^th^* process, the greater of two time values is used for the start time of this process, and then combined with the processing time of this process to obtain the completion time of the *j^th^* process for the *i^th^* workpiece.

The fitness function of the second-level evolutionary chromosome is the completion time of the workpiece and the robustness (weight sum) of scheduling scheme, and the objective function is:(8)$$f_{2}=\min\left(w\times \left(\underset{1\leq i\leq n}{\max}(C_{i})\right)+\left(1-w\right)\times r\right)$$
where *w* is the weight factor of the completion time of the workpiece and the robustness measure of scheduling scheme.

#### 3.5.3. Selection Operation

Selection simulates nature’s survival-of-fittest mechanism, where the fitter solutions survive while the weaker ones perish. The fitter chromosome receives a higher number of offspring and has a higher chance of surviving in the subsequent generation. The selection operation adopts roulette wheel selection scheme, and the chromosomes with the stronger fitness are selected to participate in the subsequent genetic operation according to the probability. The calculation of individual probability *F_i_* is determined by the fitness function, and the expression is:(9)$$F_{i}=1/f_{i}$$
where *i* = 1, 2, correspond to fitness functions of level 1 and level 2, respectively.

The population size is set as *popsize*, and the probability of this individual being selected is:(10)$$P_{i}=F_{i}/\sum_{i=1}^{popsize}F_{i}.$$

#### 3.5.4. Crossover Operation

After the selection is completed, crossover is a crucial operation in genetic algorithm. A new chromosome is generated by crossover operation for population, which promotes the population to evolve; the integer crossover method is used in crossover operation and the specific method is as follows: The two chromosomes randomly selected from the population are taken out the first *n*∗*h* bits of each chromosome, then the crossover positions randomly selected from two chromosomes are carried on the crossover; two chromosomes swap genes before and after the intersection in the first layer, the chromosome after crossing is locally adjusted, and the adjustment principle is as follows: The count of workpieces and the count of processes represented by the chromosome correspond to the constraint conditions of the scheduling problem. The chromosomal crossover operation is shown in Figure 7; the process genes of two parent chromosomes are crossed in the fourth position, the processes of some workpieces are superfluous (the first offspring: The fourth workpiece, the second offspring: The first and second workpiece), and the processes of some workpieces are missing (the first offspring: The first and second workpiece, the second offspring: The fourth workpiece); at this point, the superfluous operations of the processes are adjusted to the missing operation and the machines from the (*n*∗*h* + 1)^th^ bit to the (2∗*n*∗*h*)^th^ bit are adjusted according to the serial number of machines before the crossover (the number of machine processed the third process for the first workpiece is 1, and the number of machine processed the third process for the second workpiece is 3).

#### 3.5.5. Mutation Operation

Mutation is performed after crossover and a new chromosome can be generated through population mutation, which also promotes the whole population to evolve; the specific operation method is as follows: A mutation chromosome is selected randomly from the population, and then the mutation positions (*p*_1_ and *p*_2_) are selected; finally, the process and the corresponding machine serial number of *p*_1_ and *p*_2_ are interchanged in the mutation chromosome. The mutation operation is shown in Figure 8; the third and fifth positions of process code are interchanged in individual mutation, and the corresponding machine code are interchanged to get a new mutation chromosome.

#### 3.5.6. Double-Evolving

Double-evolving means that the evolution process of chromosomes is divided into two stages. In the first stage, the disturbance events are not considered, and the initial scheduling scheme is formed into the goal of the minimum completion time of the workpieces. In the second stage, the disturbance events are considered, the chromosome individuals formed by the evolution of the first stage are used as the initial population, and the proactive scheduling scheme is formed with the goal that the disturbance events cause the least influence (the completion time of the workpiece and the robustness measure are the least) on the initial scheduling scheme. Double-evolving aims to improve the anti-interference ability of the scheduling scheme.

Double-evolving is demonstrated by a later example in this paper. For example, six workpieces need be machined and there are three processes for each workpiece; the optional machines for each process and the processing time for every process corresponding to machines are shown in the later example. For different weight factor *w*, the different performance of the scheduling algorithm is analyzed and the result is shown in Table 1; the evolution curve of MD3GA under different weight factors is shown in Figure 9.

From Table 1 and Figure 9, when the weight factor *w* is given different values, the evolution target function value of the first stage (*f*_1_) remains unchanged in the initial scheduling scheme. When *w* = 0, the completion time of workpieces is not considered in the second-stage evolution, only the robustness measure of the scheduling scheme is considered, and at this moment, the target function value (*f*_2_ = 13.03) is minimum. When *w* = 1, the robustness measure of scheduling scheme is not considered in the second-stage evolution, only the completion time of the workpieces is considered, and at this moment, the target function value (*f*_2_ = 81) is maximum and the target function value (*f*_2_) increases linearly with the increase of *w*. Multi-object characteristics and execution performance of MD3GA are considered synthetically; *w* = 0.7 is selected in the following example.

#### 3.5.7. Double-Decoding

The initial sequence of the workpieces’ distribution is determined by the optimal chromosome gene, which is not the machining sequence of workpieces; the start time of the machining workpiece in the front gene may be later than the start time of the workpieces in the back gene. In order to solve the integrated scheduling problem of machines and single AGV, based on the strategy of first-processing and first-distributing, the double-decoding method is adopted to improve the genetic algorithm; the flow chart is shown in Figure 10. The running time of the distributing workpiece by AGV is not considered in the first decoding, and the processing sequence and machine number of each workpiece are acquired, as well as the start time and end time of machining; then, the chromosome gene is arranged in ascending order based on the start time of each process to ensure that the workpiece machined first is distributed first by AGV, and the processing sequence is not changed. The processing sequence, processing time, and distribution time are considered synthetically according to the sorted chromosomes in the second decoding; including the distribution time, the start time and end time of machining are obtained to realize the integrated scheduling of the machines and singe AGV.

All the raw materials are stored in the raw material warehouse in the second decoding, and single AGV can only take one raw material (semi-finished workpiece) to distribute every time. Before the first process of each workpiece is machined, the raw material is taken from the raw material warehouse and distributed over the corresponding machine. When the machining process is not the first process of the workpiece, the semi-finished workpiece is taken at the machine buffer of machining the previous process by AGV, and judging whether the previous process is completed, if the previous process is not completed yet, it waits to complete the process, then the semi-finished workpiece is distributed over the current machine; judging whether the previous process is completed at the current machine, if the previous process is not completed, it waits for the previous process of the current machine to be completed, and then starts the current process. Each process of all workpieces is executed in accordance with this procedure until all processes are completed, forming the specific start and end time of each process of every workpiece including AGV distribution time.

The double-decoding method is explained by the latter example in this paper. The chromosome obtained from the first decoding is sorted through ascending order according to the start processing time of the workpiece using MD3GA; the sorted chromosome is shown in Figure 11. The final time list of integrated scheduling for machines and single AGV by double-decoding is shown in Table 2, where *O_ij_* is the process for workpiece machining; start indicates the start machining time of the workpiece; end is the end time for workpiece machining; *T_W_* is a time sum, which contains the waiting time for the machine processing the previous process (the previous process is not yet completed) and the waiting time for the machine processing the current process (the previous process machined at the current machine is not yet completed); and *T_D_* is the time for distributing materials (semi-finished products) to the machines (including *T_W_*). The contents listed in the table are as follows: The processes of machining workpiece obtained from the first decoding, the process sorted through ascending order according to the start processing time, the process obtained from the second decoding including the AGV distribution time, the start processing time and the end time, and the distribution time of raw materials (semi-finished products). For example, the process (*O*_13_) is machined in machine (*M*_3_); when the semi-finished product of the first workpiece is distributed from *M*_4_ to *M*_3_, the previous process (*O*_23_) machined in *M*_3_ has not been completed. AGV needs to wait for 5 s in *M*_3_, and then *O*_13_ begins to be machined in *M*_3_.

## 4. Experimental Results and Analysis

In this section, the proactive scheduling algorithm is verified by experiment, and three key points are illustrated: (1) A real machining prototype platform is constructed; (2) system validation parameters are set; and (3) proactive scheduling is executed and results are analyzed.

### 4.1. Machining Prototype Platform

A machining prototype platform is built, which is shown in Figure 12; this platform contains four machine tools (equipped with input/output buffer), which can perform three different machining processes; an AGV equipped with mechanical arm distributes raw materials from the raw material storehouse to the machine or distributes semi-finished workpiece from the previous machine to the current machine; and only one raw material (semi-finished product) can be carried every time. Four antennas of reader (ALR-9680) are installed on the machine workstation, which is used for monitoring the abnormal events of the workpiece (paste with anti-metal tag) in real time, and the disturbance information is transmitted to the dispatch center through local area network. A wireless triaxial accelerometer is used to monitor the vibration of tools for the machining process; taking milling machine as an example, a wireless triaxial accelerometer (model: M69) is used to monitor the vibration of milling cutter, the vibration signal is transmitted to wireless base station through Zigbee wireless protocol, and then sent to the control center, with the aim of predicting the remaining useful tool life according to the vibration data. According to the abnormal events of the workpiece and the remaining useful life of the tool, the different scheduling schemes are executed to achieve the proactive scheduling of machining.

Based on Eclipse integrated development environment and MATLAB software, and embedded with rifidi-sdk3.2, Esper5.2, and other plug-ins, the man–machine interface of proactive scheduling is developed, which is shown in Figure 13, and is used to display the relevant information of the workpiece, cutter, and process during the machining process in real time. The interface includes: The display area for real-time monitoring of abnormal events of workpiece based on RFID, the display area for monitoring tool wear based on a wireless accelerometer, and the display area for proactive scheduling based on real-time abnormal events and predicted remaining useful tool life.

### 4.2. System Validation Parameters

The system configuration of scheduling algorithm programming and verification is as follows: Core i5 processor, 2.5 GHz dominant frequency, 3 GB RAM, Windows 7 (32) operating system. Scheduling algorithm is programmed using MATLAB, according to the scale and complexity of the scheduling problem, the population size is set as 500, the largest number of iterations is 500 (the number of iterations is 200 in the first-level evolution, the number of iterations is 300 in the second-level evolution), the crossover rate is 0.7, and the mutation rate is 0.3.

There are six workpieces to be machined, each workpiece is machined by three processes, every process can be machined by anyone of optional machines, and the optional machines for machining each process and the processing time on each machine have been determined. The optional machines of each process of every workpiece are shown in Table 3 and the processing time of each process of the corresponding machine is shown in Table 4.

AGV is controlled through the wireless transceiver module (433MHz) by the dispatching center. The communication protocol between the dispatching center and AGV is shown in Table 5. In this experiment, the speed of AGV is set as 2 (instruction 0001) and remains unchanged. The AGV runs in accordance with the established magnetic navigation track and has the function of magnetic detection and automatic correction. Single AGV distributes raw material (semi-finished products) to the corresponding machine during the job-shops scheduling in this paper, and only a raw material (semi-finished product) is carried every time. The layout of distributing workpiece by AGV is shown in Figure 14; O is the material/product warehouse, A, B, C, and D respectively denote machine tool *N_ncm_*_1_, *N_ncm_*_2_, *N_ncm_*_3_, and *N_ncm_*_4_, and the time matrix for running by AGV among the raw material/product warehouse and four workstations is shown in Table 6. According to different scheduling schemes (process code, machine code) formed by scheduling algorithms, the dispatching center drives AGV to complete the distribution of raw materials or semi-finished products, and AGV can automatically identify raw materials/product warehouse, workstation of each workpiece process, etc.

### 4.3. Scheduling Results and Analysis

#### 4.3.1. Dynamic Scheduling for Buffer Blocking

Firstly, the buffer blocking of a machine is monitored by RFID in real time. According to the prior analysis [26], when the processing time of each process is expanded on nine times of the initial time, the utilization rate of machine is the highest, under the conditions the dynamic scheduling experiments for buffer blocking is implemented. The initial process and machine parameters (18 processes for six workpieces are machined on four machines) are determined, the machining sequence for workpiece is optimized using MD3GA; only the completion time of workpiece machining is considered in the first evolution and forms the initial scheduling scheme; the change of the optimal solution and the population mean for the first evolution is shown in Figure 15 and the generated Gantt chart is shown in Figure 16.

It can be obtained from Figure 15 and Figure 16 that MD3GA is applied to carry out the first evolution, only the completion time of workpiece machining is taken as the optimization target, and the optimized minimum processing time (the first evolution target value) *f*_1_ = 729 s, forming the initial scheduling scheme without considering disturbance events.

Through the real-time monitoring of machining process by RFID, the system returns the state matrix S (Formula (11)) of abnormal disturbances for real-time monitoring, when the second process for the sixth workpiece is machined, the output buffer of machine tool (*M*_3_) is blocked, and the sixth workpiece occupies the machine tool; before the buffer is dredged, other processes cannot be machined on this machine temporarily; the dredging time of buffer *T_re_* = 20 s and the rescheduling strategy based on real-time event is adopted to adjust the initial scheduling scheme locally. The disturbance event for buffer blocking is considered in the second evolution; in the process of chromosome evolution, for the time of dredging machine (*M*_3_) buffer, the right shift strategy of machining process of *M*_3_ is adopted; that is for each chromosome in the population, other processes shift right the time of *T_re_* after the completion of the second process for the sixth workpiece machined on *M*_3_. Minimizing the completion time of the workpiece and weight sum of robustness measure is the target to optimize, aiming to minimize the impact on the initial scheduling scheme for real-time event of buffer blocking. The change of optimal solution and population mean in the second evolution is shown in Figure 17 and the Gantt chart of dynamic scheduling scheme for buffer blocking is shown in Figure 18.

The state matrix of abnormal events monitored by RFID is as follows:(11)$$S=\left[\begin{array}{cccc} 1 & 1 & 1 & 1\\ 1 & 1 & 1 & 1\\ 1 & 1 & 1 & 1\\ 1 & 1 & 1 & 1\\ 1 & 1 & 1 & 1\\ 1 & 1 & 0 & 1 \end{array}\right].$$

The optimal machining sequence has been generated through the first evolution of Figure 17 and the real-time disturbance event of buffer blocking is responded to. Minimizing the completion time of workpiece and weight sum of robustness measure is the target to adjust the initial scheduling scheme locally, obtaining the optimal target *f*_2_ = 519.12, with the aim of minimizing the impact of the initial scheduling scheme for disturbance event. By comparing Figure 16 and Figure 18, in order to deal with buffer blocking event occurring in machining *O*_62_, only processes *O*_13_ and *O*_53_ after *O*_62_ are shifted right the time of *T_re_*, and other processes remain unchanged, avoiding the impact on processes *O*_13_ and *O*_53_ for disturbance event of buffer blocking, and ensuring the minimum impact on the initial scheduling scheme.

#### 4.3.2. Proactive Scheduling Based on Tool Wear Prediction

Secondly, the case of monitoring tool wear by a wireless accelerometer and predicting the remaining useful life is discussed. Initial conditions are the same as dynamic scheduling part; the machining sequence of the workpiece is optimized using MD3GA, only the completion time of workpiece machining is considered in the first evolution, as well as forming the initial scheduling scheme. The change of the optimal solution and the population mean for the first evolution is shown in Figure 19, the generated Gantt chart is shown in Figure 20, and the Gantt chart of integration scheduling including AGV distribution time is shown in Figure 21.

It can be obtained from Figure 19 and Figure 20 that MD3GA is applied to carry out the first evolution; only the completion time of workpiece machining is taken as the optimization target and the optimized minimum processing time (the first evolution target value) *f*_1_ = 747 s, forming the initial scheduling scheme without considering disturbance events.

The vibration is monitored by a wireless triaxial accelerometer during the process of machining, and predicting remaining useful life for tool, the remaining useful life matrix (the remaining useful life is expressed by the number of workpieces machined with this tool also) is obtained; when the initial scheduling scheme in not performed, the tool for *M*_3_ is also predicted for the machine 2 processes; namely, *W*_3_ = 2 is expressed in the remaining useful life matrix (Formula (12)); this tool needs to be replaced after machining 2 processes to machine the latter processes again. The time for replacing tool *T_c_* = 30 s, the rescheduling strategy based on predicted event is adopted to perform the full rescheduling scheme, and the predicted event of exhausting tool life is considered in the second evolution. In the process of chromosome evolution, for the time of replacing tool on *M*_3_, the shift right strategy of machining processes machined on *M*_3_ is also adopted; namely, for each chromosome in the population, the start time and end time of other processes shift right the time of *T_c_* after two processes are machined on *M*_3_. Minimizing the completion time of workpiece and weight sum of robustness measure is the target to optimize, with the aim of minimizing the impact of the initial scheduling scheme for the predicted event of exhausting tool life. The change of optimal solution and population mean in the second evolution is shown in Figure 22, the Gantt chart of proactive scheduling scheme for tool wear prediction is shown in Figure 23, and the Gantt chart of integration scheduling including AGV distribution time is shown in Figure 24.

The wireless accelerometer is used to monitor the vibration of the tool and predict the remaining useful life. The matrix of remaining useful life for tool is as follows:(12)$$W=\left[\begin{array}{cccc} 50 & 100 & 2 & 150 \end{array}\right].$$

Before performing the initial scheduling scheme, it was predicted that only two processes can be machined in *M*_3_, and the tool life will be exhausted and the tool would need to be replaced in order to deal with the predicted event; the machined workpieces performed the full rescheduling scheme, with the optimal target *f*_2_ = 532.36 in the second evolution. By comparing Figure 20 and Figure 23, in order to deal with the predicted disturbance event of replacing tool after machining *O*_12_, the processes *O*_13_ and *O*_53_ after *O*_12_ machined in *M*_3_ are shifted right the time of *T_c_* in the second evolution. Minimizing the completion time of workpiece and weight sum of robustness measure is the target; the machining sequence is sorted again, with the aim of minimizing the impact of the initial scheduling scheme for the disturbance event of replacing tool; process *O*_13_ is still machined in *M*_3_, which is after process *O*_12_ (the initial scheme is shifted right the time of *T_c_*). *O*_23_ is machined in *M*_3_, *O*_53_ is machined on *M*_1_, *O*_33_ machined in *M*_1_ in the initial scheduling scheme is arranged to machine in *M*_4_, avoiding the impact on *O*_13_ and *O*_53_ for the disturbance event of replacing tool, and at the same time, to ensure the minimum impact on the initial scheduling scheme. The machining processes have been adjusted in Figure 24. The actual machining time of workpieces is 756 s, 9 s more than the initial scheduling scheme; the AGV running time is 231 s, the AGV waiting time is 421 s, and the AGV waiting time is greater than its running time due to the workpiece distribution by single AGV.

## 5. Conclusions and Future Work

Proactive scheduling based on the abnormal event monitoring of the workpiece and remaining useful life prediction of the tool is proposed and verified in this paper. Based on the real-time abnormal events of the workpiece and remaining useful life of the tool, the hybrid rescheduling strategy based on real-time event and predicted event is adopted; the completion time of workpieces and the robustness measure of scheduling scheme are considered synthetically; MD3GA is proposed, with the aim of minimizing the impact of the initial scheduling scheme for abnormal disturbances; the prototype platform for machining workpiece is constructed; a human–machine interface is developed, to realize the integration scheduling of machines and single AGV; and the feasibility of proactive scheduling is verified.

When exploring the proactive scheduling method, a prototype platform system is constructed to verify the method in this paper, and a lot of constraints are added during the execution, such as AGV running along the magnetic orbit, etc. In future research on proactive scheduling, AGV path planning and collaborative optimization of the robotic arm will be considered at the same time. We only take into account the RFID and accelerometer data in the method validation, human factors are not involved, and the amount of data collected is limited; the experimental environment is not as complex as actual workshop site. In real situations, the following factors should have been considered: The environmental factors, the vast amounts of different types of data collection, the more complex machining processes, more equipment, as well as the uncertainty of the workers, etc.

## Figures and Tables

**Figure 1 sensors-19-05254-f001:**
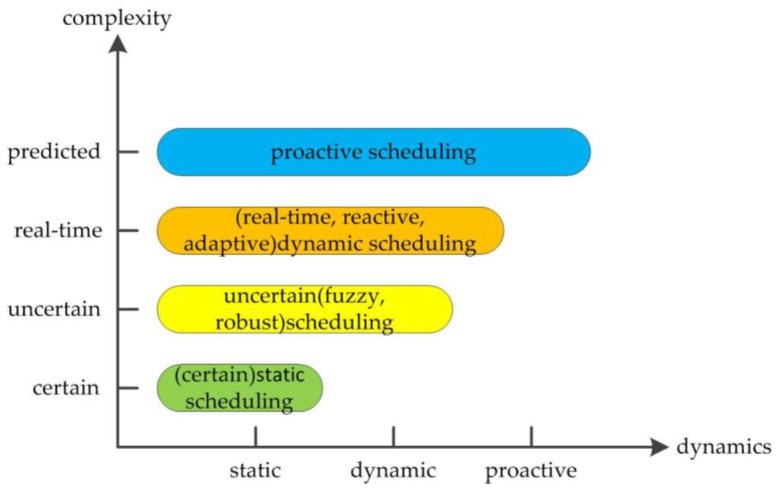
Classification of production scheduling models.

**Figure 2 sensors-19-05254-f002:**
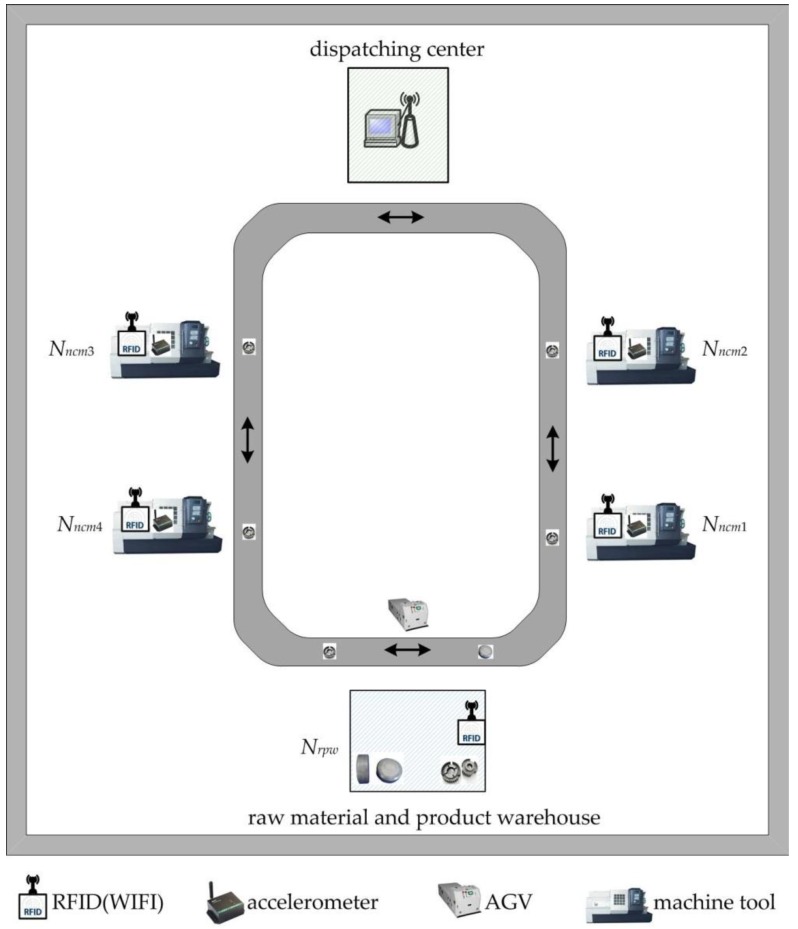
A sensing-aware environment of machining job in wisdom workshop.

**Figure 3 sensors-19-05254-f003:**
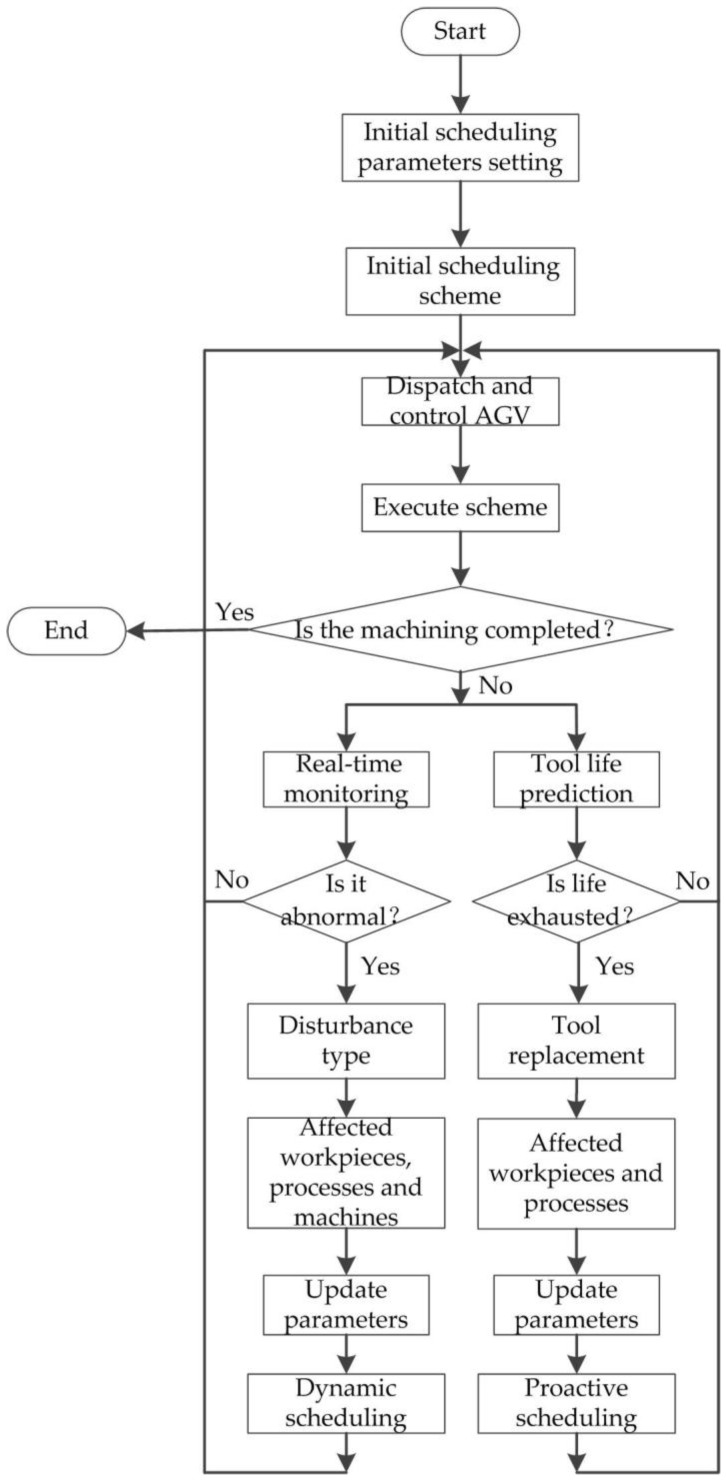
A dynamic and proactive scheduling framework of machining jobs.

**Figure 4 sensors-19-05254-f004:**
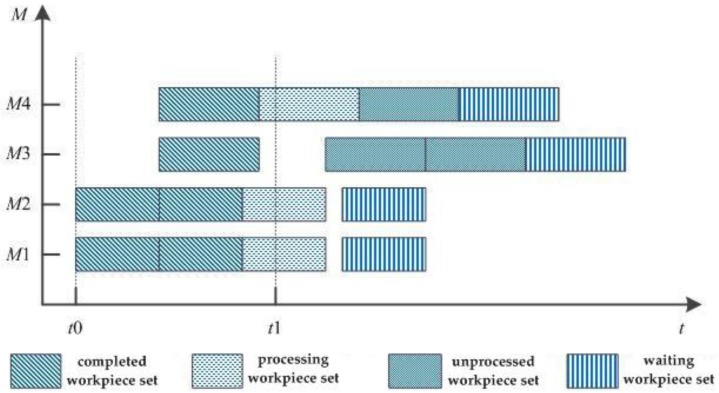
A workpiece window of rolling scheduling.

**Figure 5 sensors-19-05254-f005:**
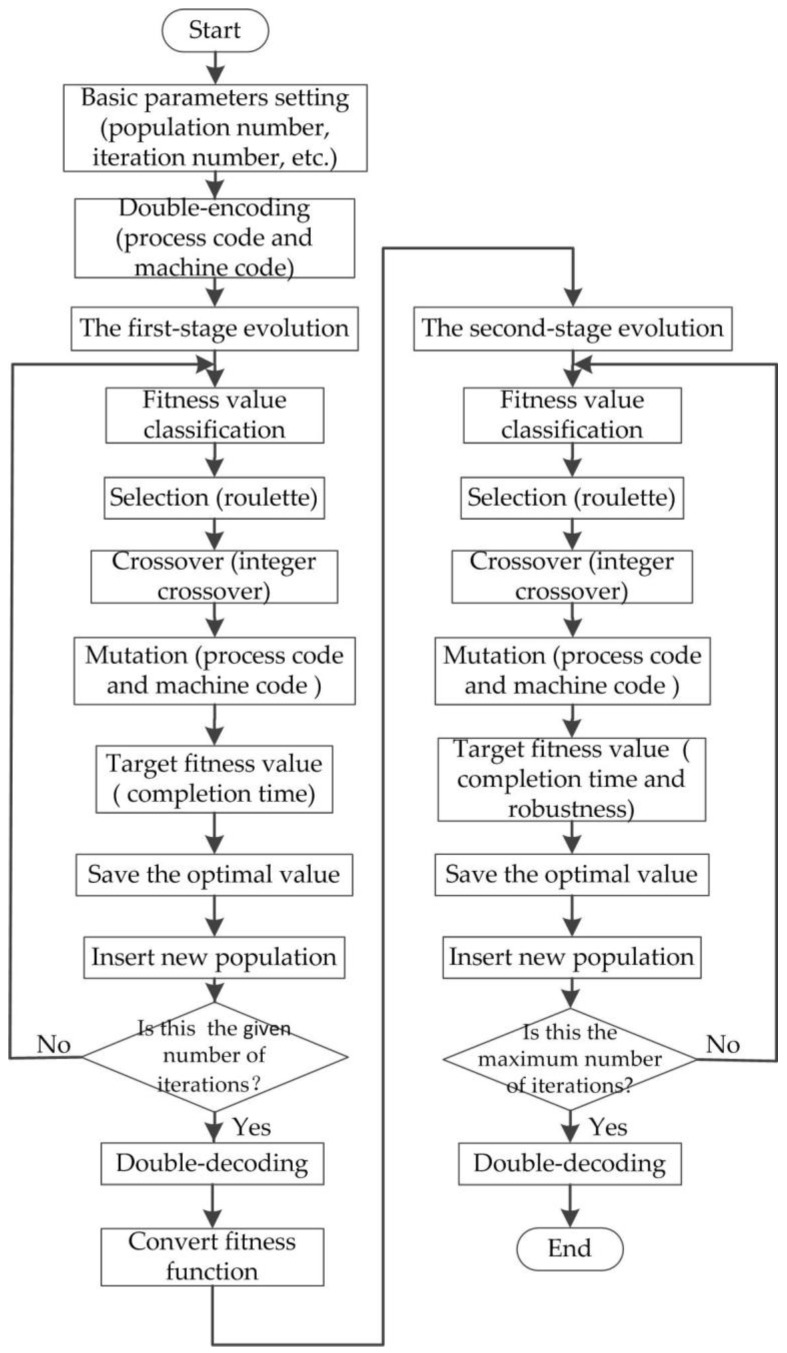
A flow chart of the multi-objective, double-encoding, double-evolving, and double-decoding genetic algorithm (MD3GA).

**Figure 6 sensors-19-05254-f006:**
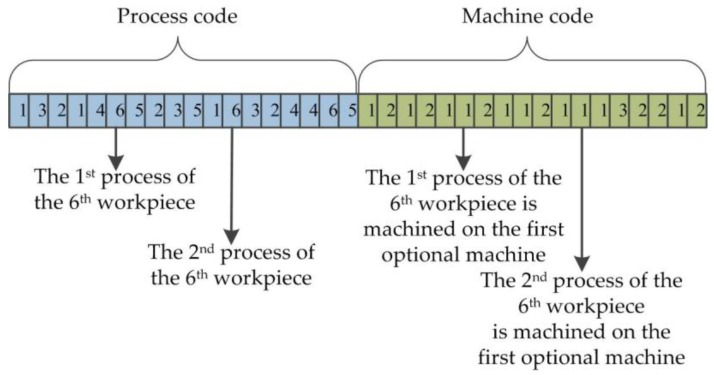
The chromosome double-encoding.

**Figure 7 sensors-19-05254-f007:**
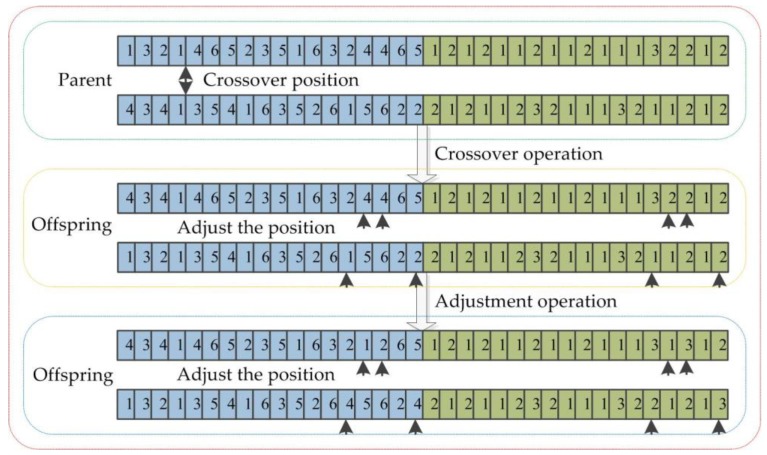
The chromosome crossover.

**Figure 8 sensors-19-05254-f008:**
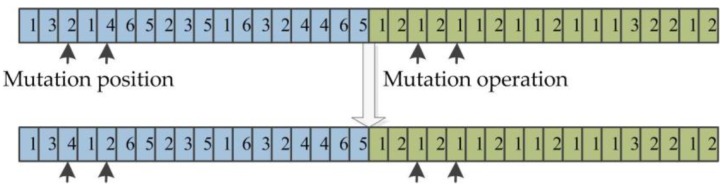
The chromosome mutation.

**Figure 9 sensors-19-05254-f009:**
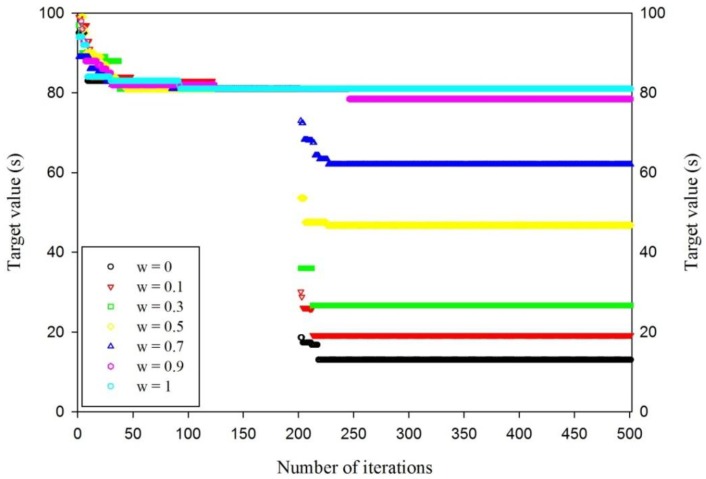
The evolution curve of MD3GA under different weight factors.

**Figure 10 sensors-19-05254-f010:**
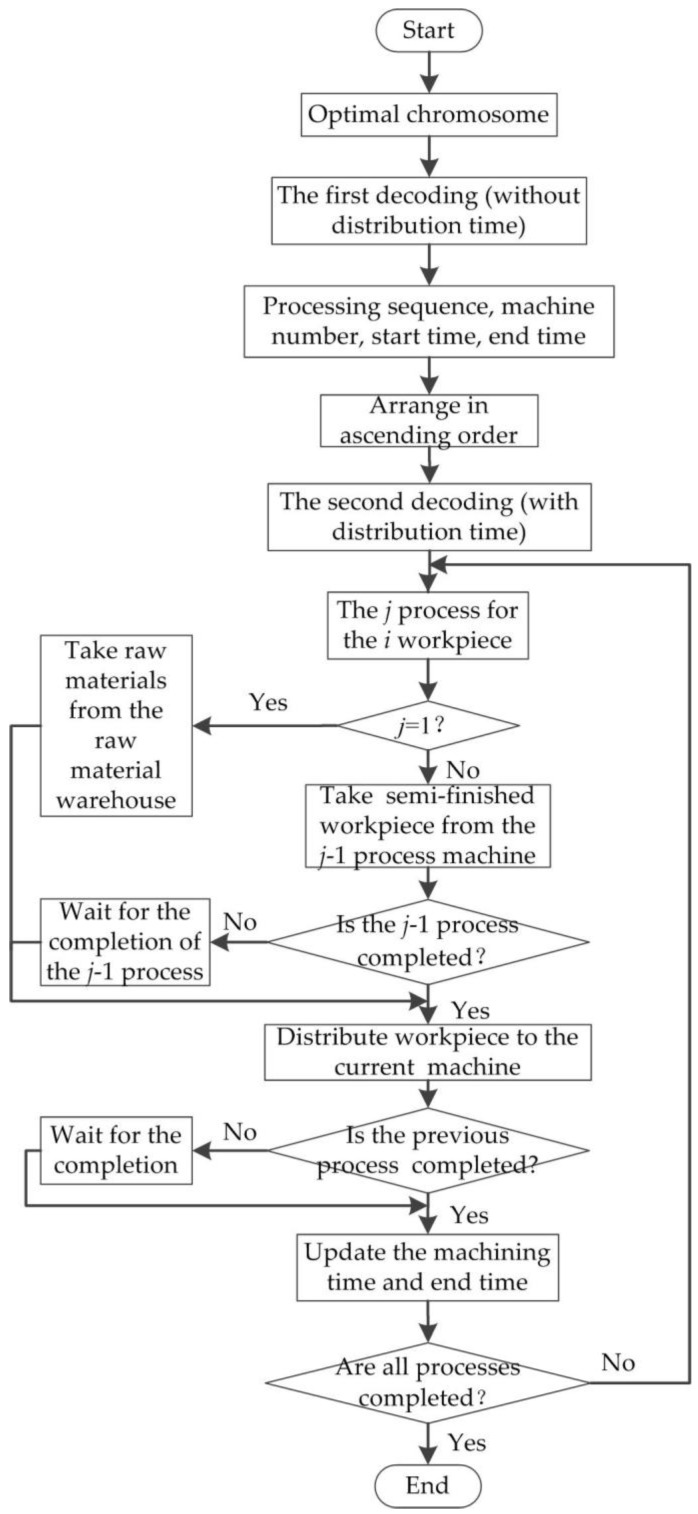
A flow chart of double decoding.

**Figure 11 sensors-19-05254-f011:**
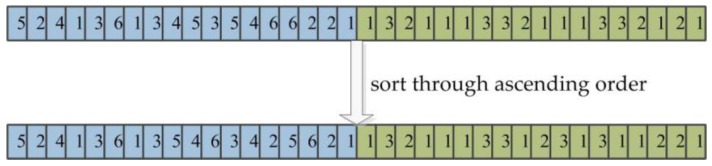
An ascending sort of a chromosome according to start processing time.

**Figure 12 sensors-19-05254-f012:**
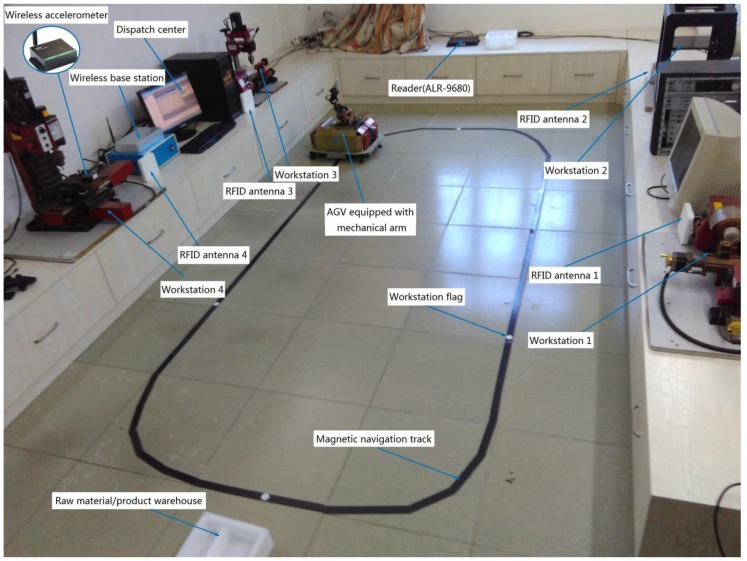
A prototype platform of machining jobs.

**Figure 13 sensors-19-05254-f013:**
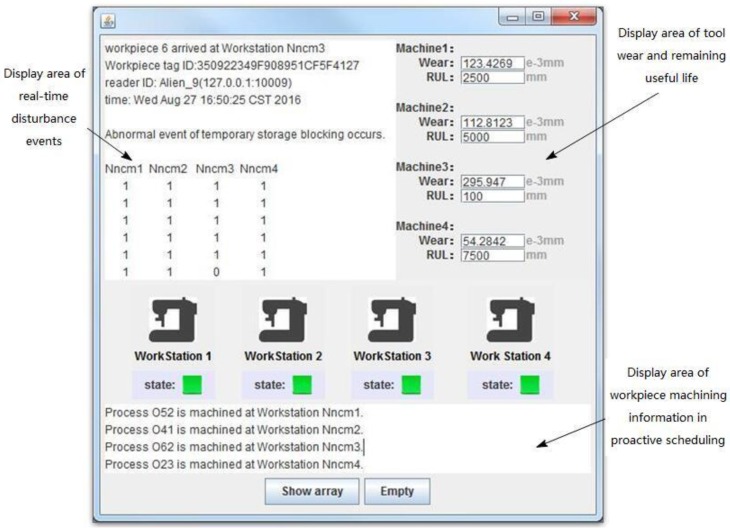
Human–machine interface of the prototype platform for proactive scheduling.

**Figure 14 sensors-19-05254-f014:**
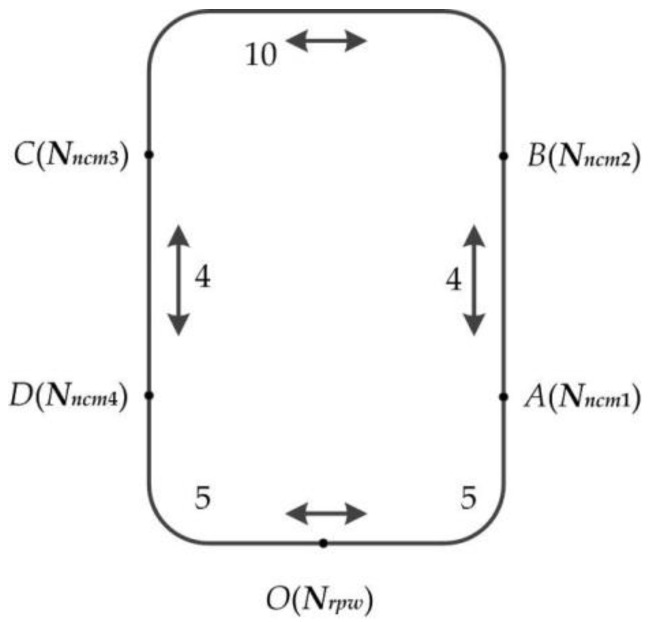
The layout of dispatching workpieces by the AGV.

**Figure 15 sensors-19-05254-f015:**
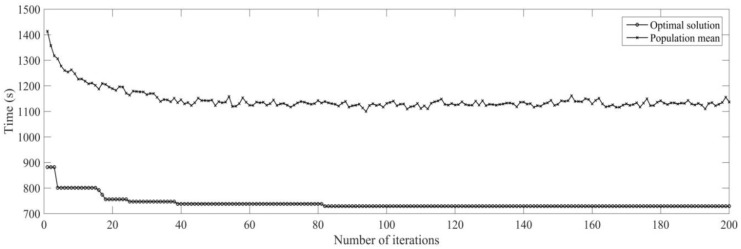
The change of the optimal solution and the population mean in the first evolution.

**Figure 16 sensors-19-05254-f016:**
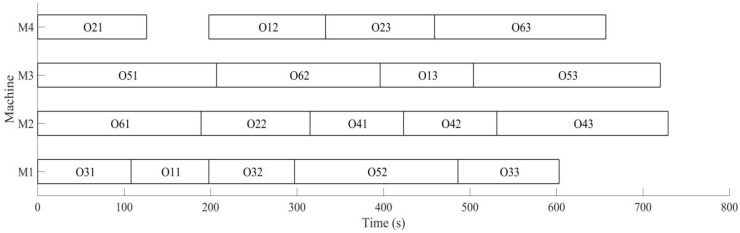
A Gantt chart of the initial scheduling scheme in the first evolution.

**Figure 17 sensors-19-05254-f017:**
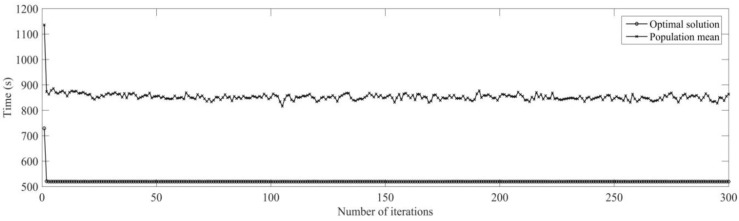
The change of the optimal solution and the population mean in the second evolution.

**Figure 18 sensors-19-05254-f018:**
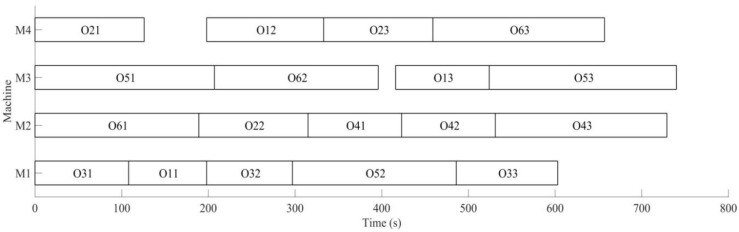
A Gantt chart of dynamic scheduling for buffer blocking in the second evolution.

**Figure 19 sensors-19-05254-f019:**
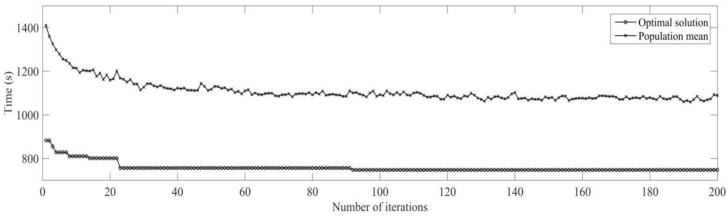
The change of the optimal solution and the population mean in the first evolution.

**Figure 20 sensors-19-05254-f020:**
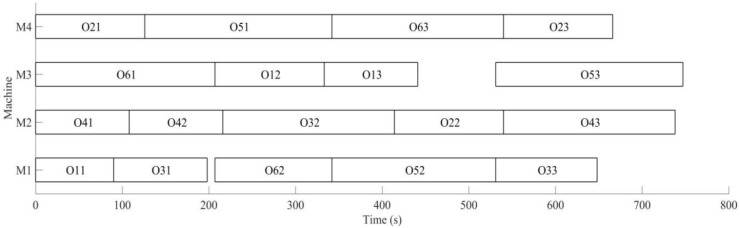
A Gantt chart of the initial scheduling scheme in the first evolution.

**Figure 21 sensors-19-05254-f021:**
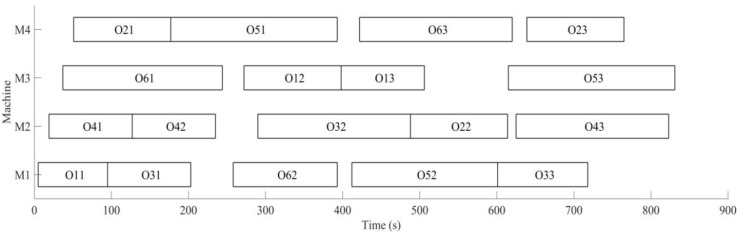
A Gantt chart of the initial scheduling scheme including dispatching time of the AGV.

**Figure 22 sensors-19-05254-f022:**
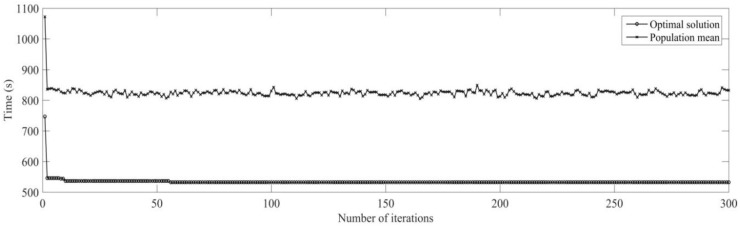
The change of the optimal solution and the population mean in the second evolution.

**Figure 23 sensors-19-05254-f023:**
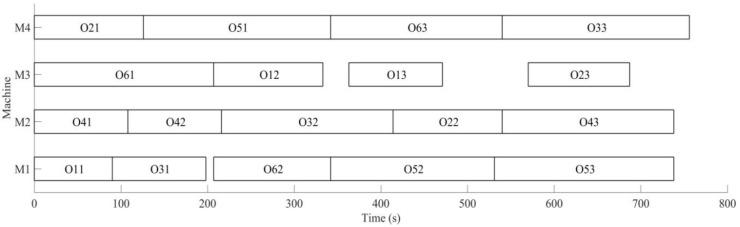
A Gantt chart of proactive scheduling based on tool wear prediction in the second evolution.

**Figure 24 sensors-19-05254-f024:**
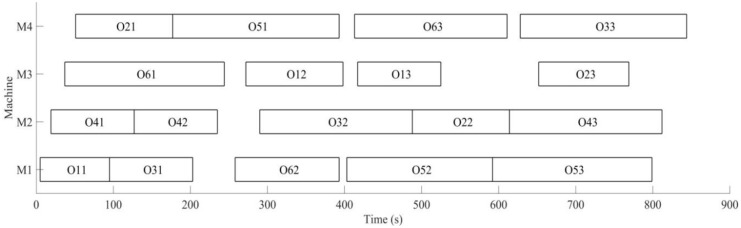
A Gantt chart of proactive scheduling including dispatching time of the AGV.

**Table 1 sensors-19-05254-t001:** The performance of MD3GA under different weight factors.

Weight Factor (*w*)	Target (*f*_1_)	Target (*f*_2_)	Run Time (s)
0	81	13.03	352
0.1	81	19.21	481
0.3	81	26.67	518
0.5	81	46.81	418
0.7	81	62.06	412
0.9	81	78.46	536
1.0	81	81	407

**Table 2 sensors-19-05254-t002:** A time list of double-decoding.

Item	Process (s)																		
The first decoding	*O_ij_*	*O* _51_	*O* _21_	*O* _41_	*O* _11_	*O* _31_	*O* _61_	*O* _12_	*O* _32_	*O* _42_	*O* _52_	*O* _33_	*O* _53_	*O* _43_	*O* _62_	*O* _63_	*O* _22_	*O* _23_	*O* _13_
	Start	0	0	0	0	10	12	14	23	33	23	44	57	45	36	59	45	59	72
	End	23	14	12	10	22	33	29	36	45	44	57	80	59	57	81	59	72	84
Sort in ascending order	*O_ij_*	*O* _51_	*O* _21_	*O* _41_	*O* _11_	*O* _31_	*O* _61_	*O* _12_	*O* _32_	*O* _52_	*O* _42_	*O* _62_	*O* _33_	*O* _43_	*O* _22_	*O* _53_	*O* _63_	*O* _23_	*O* _13_
	Start	0	0	0	0	10	12	14	23	23	33	36	44	45	45	57	59	59	72
	End	23	14	12	10	22	33	29	36	44	45	57	57	59	59	80	81	72	84
The second decoding	*O_ij_*	*O* _51_	*O* _21_	*O* _41_	*O* _11_	*O* _31_	*O* _61_	*O* _12_	*O* _32_	*O* _52_	*O* _42_	*O* _62_	*O* _33_	*O* _43_	*O* _22_	*O* _53_	*O* _63_	*O* _23_	*O* _13_
	Start	9	23	37	51	61	75	89	113	127	131	141	155	173	187	191	209	233	246
	End	32	37	49	61	73	96	104	126	148	143	162	168	187	201	214	231	246	258
Waiting time	*T_W_*	0	0	0	0	0	0	0	0	0	0	0	0	0	0	0	0	0	5
Distribution time	*T_D_*	9	14	14	14	10	14	14	24	14	4	10	14	18	14	4	18	24	13

**Table 3 sensors-19-05254-t003:** An optional machine list of machining processes.

Workpiece	Process 1	Process 2	Process 3
*W* _1_	(1, 2, 3)	(2, 3, 4)	(3, 4)
*W* _2_	(2, 3, 4)	(2, 3)	(1, 3, 4)
*W* _3_	(1, 3, 4)	(1, 2, 3)	(1, 4)
*W* _4_	(1, 2, 3)	(1, 2)	(1, 2, 4)
*W* _5_	(3, 4)	(1, 3)	(1, 3, 4)
*W* _6_	(2, 3)	(1, 2, 3)	(1, 4)

**Table 4 sensors-19-05254-t004:** A processing time list of machining processes on the corresponding machines.

Workpiece	Process 1 (s)	Process 2 (s)	Process 3 (s)
*W* _1_	(10, 15, 20)	(24, 14, 15)	(12, 24)
*W* _2_	(24, 27, 14)	(14, 23)	(23, 13, 14)
*W* _3_	(12, 13, 24)	(11, 22, 13)	(13, 24)
*W* _4_	(21, 12, 23)	(21, 12)	(21, 22, 14)
*W* _5_	(23, 24)	(21, 23)	(23, 24, 26)
*W* _6_	(21, 23)	(15, 17, 21)	(21, 22)

**Table 5 sensors-19-05254-t005:** Communication protocol between the dispatching center and the automatic guided vehicle (AGV).

Action	Instruction	Action	Instruction
Start	8000	Turn left	0040
Stop	0020	Turn right	0080
Speed 1	0000	Rotate 90° clockwise	0100
Speed 2	0001	Rotate 90° anticlockwise	0200
Speed 3	0002	Rotate 180° clockwise	0500
Speed 4	0003	Rotate 180° anticlockwise	0600

**Table 6 sensors-19-05254-t006:** A time matrix of running between two workstations for the AGV.

Workstation	*O* (s)	*A* (s)	*B* (s)	*C* (s)	*D* (s)
*O* (s)	0	5	9	9	5
*A* (s)	5	0	4	14	10
*B* (s)	9	4	0	10	14
*C* (s)	9	14	10	0	4
*D* (s)	5	10	14	4	0

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
