# Peer review of "Proactive Scheduling for Job-Shop Based on Abnormal Event Monitoring of Workpieces and Remaining Useful Life Prediction of Tools in Wisdom Manufacturing Workshop"

_sensors, 2019, doi:10.3390/s19235254_

Round 1

Reviewer 1 Report

The authors did not address the comments from all reviewers. Please revise your manuscript accordingly. 

Thanks

Author Response

Response to Reviewer 1 Comments

Point 1: The authors did not address the comments from all reviewers. Please revise your manuscript accordingly.  

Response 1: The following comments were mentioned in the previous comments: “Please also review relevant work (such as works at the University of Alberta as below) in the literature review, which could be helpful in defining your contribution”. We review some literatures about the works at the University of Alberta, which are summarized as follows: “Rokni et al. adopted the Pareto-optimality concept combined with fuzzy set theory to optimize the pipe spool fabrication shop scheduling [1]. Taghaddos et al. proposed the Simulation-Based Auction Protocol (SBAP) to realize the effective allocation of resources and satisfaction of various constraints [2]”. The contribution of this paper is summarized clearly, which is marked using the "Track Changes" function in Section 2.

Rokni, S.; Fayek, A.R. A multi-criteria optimization framework for industrial shop scheduling using fuzzy set theory. Integrated Computer-Aided Engineering 2010, 17, 175-196, doi:10.3233/ica-2010-0344. Taghaddos, H.; Hermann, U.; AbouRizk, S.; Mohamed, Y. Simulation-Based Multiagent Approach for Scheduling Modular Construction. Journal of Computing in Civil Engineering 2014, 28, 263-274, doi:10.1061/(asce)cp.1943-5487.0000262.

Reviewer 2 Report

I notice the solid methodology in approaching the problem: from identifying the problem, developing the mathematical model, proposing the solution algorithm and until imagining an experiment to validate the method. Congratulations on this beautiful work!

A question for the authors: could they correlate their method with the better known Petri nets, used for many years in the methodology of programming, monitoring and control in the industry?

Author Response

Response to Reviewer 2 Comments

Point 1: A question for the authors: could they correlate their method with the better known Petri nets, used for many years in the methodology of programming, monitoring and control in the industry? 

Response 1: Your comment is a good suggestion. Recently, I also reviewed some literatures about Petri nets, which are used for deadlock control of automated manufacturing systems by the principles and techniques that are involved in preventing, avoiding, and detecting deadlocks [1,2]. A novel and computationally efficient method to design optimal control places, and an iteration approach that only computes the reachability graph of a plant Petri net model are adopted to obtain a maximally permissive liveness-enforcing supervisor for a flexible manufacturing system [3]. A Petri net approach to modeling, analysis, simulation, scheduling, and control is adopted in a semiconductor manufacturing systems [4]. The Petri nets and a uniform environment for modelling, formal analysis, and design of discrete event systems are introduced in industrial applications [5]. Petri net combined with heuristic search is adopted to formulate a scheduling problem in flexible manufacturing system [6]. There are some abnormal events about workpiece and equipment to be monitored in our works, these discrete events that exhibit sequential, concurrent, and conflicting relations among the events and operations, which is dynamic over time. These events and their relationships can be characterized to perform the scheduling by Petri nets. We try to apply Petri nets to our works in the future.

Li, Z.; Wu, N.; Zhou, M. Deadlock Control of Automated Manufacturing Systems Based on Petri Nets-A Literature Review. Ieee Transactions on Systems Man and Cybernetics Part C-Applications and Reviews 2012, 42, 437-462, doi:10.1109/tsmcc.2011.2160626. Ezpeleta, J.; Colom, J.M.; Martinez, J. A Petri net based deadlock prevention policy for flexible manufacturing systems. IEEE Trans. Robot. Autom. 1995, 11, 173-184, doi:10.1109/70.370500. Chen, Y.; Li, Z.; Khalgui, M.; Mosbahi, O. Design of a Maximally Permissive Liveness-Enforcing Petri Net Supervisor for Flexible Manufacturing Systems. IEEE Trans. Autom. Sci. Eng. 2011, 8, 374-393, doi:10.1109/tase.2010.2060332. Zhou, M.C.; Jeng, M.D. Modeling, analysis, simulation, scheduling, and control of semiconductor manufacturing systems: A Petri net approach. IEEE Trans. Semicond. Manuf. 1998, 11, 333-357, doi:10.1109/66.705370. Zurawski, R.; MengChu, Z. Petri nets and industrial applications: A tutorial. IEEE Trans. Ind. Electron. 1994, 41, 567-583, doi:10.1109/41.334574. Doo Yong, L.; DiCesare, F. Scheduling flexible manufacturing systems using Petri nets and heuristic search. IEEE Trans. Robot. Autom. 1994, 10, 123-132, doi:10.1109/70.282537.

Reviewer 3 Report

This study presents a proactive job-shop scheduling method based om abnormal event monitoring of workpieces and remaining useful life prediction of tools. The study considered dynamic scheduling in the actual environment of manufacturing, that is often ignored in literature. An actual prototype platform was built to verify the proposed method. 

Overall, the paper is well-prepared. A critical literature review was carried out to emphasize the unique of this study. The method and the experiment were described in details. The experimental results verified the feasibility of the scheduling method. The paper can be accepted for publication after addressing the following comments. 1. Several abbreviations were not explained for the first use. 2. What is the limitation of the proposed method when implemented in real situations? and please add it to the conclusion part.

Author Response

Response to Reviewer 3 Comments

Point 1: Several abbreviations were not explained for the first use. 

Response 1: Some abbreviations are explained in the revised manuscript. For example, Radio Frequency IDentification (RFID), Teaching Learning (TL), Local Area Network (LAN), Wireless Fidelity (WiFi), IDentification (ID), which are marked using the "Track Changes" function.

Point 2: What is the limitation of the proposed method when implemented in real situations? and please add it to the conclusion part.

Response 2: We only takes into account the RFID and accelerometer data in the method validation, human factors is not involved, and the amount of data collected is limited, the experimental environment is not as complex as actual workshop site. In real situations, the following factors should been considered: the environmental factors, the vast amounts of different types of data collection, the more complex machining processes, the more equipment as well as the uncertainty of the workers, etc., which is marked using the "Track Changes" function in Section 5.

Round 2

Reviewer 1 Report

Thanks for addressing my comments. 

This manuscript is a resubmission of an earlier submission. The following is a list of the peer review reports and author responses from that submission.

Round 1

Reviewer 1 Report

In this paper, the authors proposed a proactive scheduling based on the abnormal event monitoring of the workpiece and remaining useful life prediction of the tool. An experiment with 6 workpieces, 3 processes, 4 machines, and 1 AGV is employed to verify the effectiveness of the proposed method. To improve the readability of this paper, the following issues should be considered.

The grammatical and typo errors should be corrected at first. The genetic operators, initialization, selection, crossover, mutation, and elimination, of MD3GA need to be described in more detail. How to define the “Remaining Useful Life Prediction of Tools” in this paper? The descriptions of Fig. 6 seems incorrect. On the right side, “The 1st process is …” should be corrected to “The 1st process of the 6th workpiece …”.

    5. On line 320, the authors said that “6 workpieces on 4 machines …” but there are only 3 machines shown in Machine code.

Reviewer 2 Report

This manuscript presented proactive scheduling based on abnormal event monitoring of workpieces and remaining useful life prediction of tools. Overall, the paper is well-written and is back up with solid research. 

One primary concern is that the authors presented too many things, such as proactive scheduling, RFID and wireless accelerometer, Proactive Scheduling Mathematical Model, Proactive Scheduling Framework, Proactive Scheduling Strategy, Proactive Scheduling Algorithm; then which one is your main scientific contribution?

Most of the literature review is related to scheduling optimization algorithms. So, your main contribution is proactive scheduling algorithm? Genetic algorithm has been well studied, then, what is the unique strength of your proactive scheduling algorithm?

Please explicitly clarify your contributions and the questions above in your revised manuscript.

Also, many studies have been conducted to use simulation and/or optimization for dynamic scheduling in consideration of uncertainties. Please also review relevant work (such as works at the University of Alberta as below) in the literature review, which could be helpful in defining your contribution. 

Reviewer 3 Report

Authors propose the two stages of genetic algorithm for Job-Shop Scheduling problem.

According to the proposed flowchart of GA,

local optimal may happen; some proposed multiple-objective genetic algorithms already can find the optimal solutions by adaptive crossover method instead of two independent stages genetic algorithm multiple-objective genetic algorithms may cause the pareto-front solutions. To use weight sum method for reducing the number of objects may not be acceptable or reasonable.